# GraphCLIP: Enhancing Transferability in Graph Foundation Models for Text-Attributed Graphs

## Abstract

Recently, research on Text-Attributed Graphs (TAGs) has gained significant attention due to the prevalence of free-text node features in real-world applications and the advancements in Large Language Models (LLMs) that bolster TAG methodologies. However, current TAG approaches face two primary challenges: (i) *Heavy reliance on label information* and (ii) *Limited cross-domain zero/few-shot transferability*. These issues constrain the scaling of both data and model size, owing to high labor costs and scaling laws, complicating the development of graph foundation models with strong transferability. In this work, we propose the GraphCLIP framework to address these challenges by learning graph foundation models with strong cross-domain zero/few-shot transferability through a self-supervised contrastive graph-summary pretraining method. Specifically, we generate and curate large-scale graph-summary pair data with the assistance of LLMs, and introduce a novel graph-summary pretraining method, combined with invariant learning, to enhance graph foundation models with strong cross-domain zero-shot transferability. For few-shot learning, we propose a novel graph prompt tuning technique aligned with our pretraining objective to mitigate catastrophic forgetting and minimize learning costs. Extensive experiments show the superiority of GraphCLIP in both zero-shot and few-shot settings, while evaluations across various downstream tasks confirm the versatility of GraphCLIP. Our code is available at: https://anonymous.4open.science/r/GraphCLIP

## 1 Introduction

Text-Attributed Graphs (TAGs) have gained significant attention recently [5, 9, 14, 30, 43, 63, 68, 79] due to the free-text node feature space prevalent in various domains such as social, e-commerce, and citation networks [6, 26, 58]. TAGs offer two natural advantages for graph learning research: (i) all node features can be aligned into within the same text space, enabling the model to transfer effectively across different graphs, and (ii) powerful off-the-shelf tools can be readily leveraged to address challenges within TAGs, *e.g.*, Large Language Models (LLMs) can be used for enriching the textual representations of TAGs.

**Existing TAG methods with LLMs.** Significant efforts are underway to combine TAGs with LLMs, aiming to develop Graph Foundation Models (GFMs) [4, 21, 30, 31, 33, 53], a promising approach that enables the transfer of knowledge from source data to tackle various downstream tasks on target data through a unified backbone. Existing methods can be categorized into three main types [21, 30]: LLM as Enhancer, LLM as Predictor, and LLM as Aligner, as illustrated in Figure 1.

LLM as Enhancer involves leveraging language models to augment raw text, yielding refined, high-quality outputs, or to encode node features, surpassing previous shallow embedding methods like Bag of Words (BoW). For instance, OFA [33] and ZeroG [31]

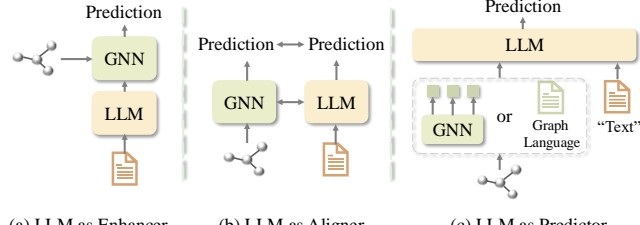

(a) LLM as Enhancer    (b) LLM as Aligner    (c) LLM as Predictor

**Figure 1: Three main categories of TAG methods.**

utilize language models as node feature extractors, employing labeled source data to pretrain a graph model, which is then applied to target data using complicated graph prompt techniques. These methods *depend heavily on high-quality labeled data*, potentially limiting the full potential of graph foundation models due to scaling laws [15, 22]. The challenge arises from the difficulty of scaling up the pretraining corpus, primarily due to the associated high labor costs. In the paradigm of LLM as Predictor, the most essential task is to map graph data to a format that LLMs can comprehend. For example, GraphGPT [53] and LLaGA [4] utilize GNNs to encode graph data into graph tokens, training an additional projector to map these tokens into the text space through instruction tuning. However, these methods also *require high-quality labels* for target data, and recent studies [5, 32] have shown they *exhibit poor cross-domain zero-shot performance*. LLM as Aligner involves mapping graph and text modalities into a shared embedding space. For example, ConGrat [3] and G2P2 [64] apply self-supervised graph-text contrastive pretraining and focus on transferring pretrained models within the *same* graph, neglecting the cross-domain or cross-graph transferability of these models.

**Challenges for current TAG methods.** As summarized, current methodologies encounter two primary challenges:

(i) *Heavy reliance on label information*: Most current approaches such as ZeroG [31], OFA [33], LLaGA [4], and others [14, 53, 63, 79] require label information from source data as training signals, leading to significant labor costs. This constraint prevents GFMs from leveraging extensive unlabeled training corpora, thereby restricting the scaling of both data and model sizes in accordance with scaling laws [15].

(ii) *Limited cross-domain zero/few-shot transferability*: A well-pretrained GFM should be applicable directly to target data, achieving satisfactory performance without any parameter adjustments, *i.e.*, strong cross-domain/dataset zero-shot capability, like CLIP [44] in multi-modal domain and LLMs in NLP domain. However, most existing methods struggle with direct deployment in zero-shot settings on target data. They either cannot perform zero-shot learning because they require the training of a classification head using labeled target data to generate predictions [14, 72, 79], or they

demonstrate inadequate zero-shot performance due to insufficient transferable knowledge acquired during pretraining [4, 33]. Additionally, in low-resource scenarios like few-shot learning, effectively leveraging limited training samples from target data while circumventing catastrophic forgetting poses a significant challenge.

**Our proposed GraphCLIP framework.** We develop GraphCLIP to address the challenges above, which learns graph foundation models with robust cross-domain zero-shot transferability from a novel self-supervised contrastive graph-summary pretraining technique. Specifically, we leverage the open LLM, QWen2-72B [69], to generate graph-related summary texts of subgraphs, capitalizing on their powerful summarization abilities. Utilizing generated large-scale graph-summary pairs, we train a cross-domain graph foundation model through designed self-supervised contrastive graph-summary pretraining (addressing *Challenge (i)*). Considering the necessity of out-of-domain generalization across graphs, we introduce invariant learning during pretraining to capture invariant features, thereby enhancing out-of-domain generalization. After pretraining, GraphCLIP can be applied directly to target data without fine-tuning, demonstrating strong zero-shot capability on both in-domain and cross-domain graph data. For few-shot scenarios, we propose a novel graph prompt tuning approach aligned with our pretraining objective, reducing catastrophic forgetting [49] and minimizing learning costs [36, 50], ensuring excellent performance in few-shot settings (addressing *Challenge (ii)*). Extensive experiments show that GraphCLIP exhibits strong zero-shot performance across various in-domain and cross-domain target datasets. In few-shot settings, GraphCLIP with our designed prompt tuning outperforms previous state-of-the-art methods. Additionally, the universality of our method is demonstrated through evaluation across various downstream tasks.

Our contributions can be concluded as:

- We generate and curate a large-scale graph-summary pair dataset which contains over 0.2B tokens with the assistance of LLMs, establishing a valuable training corpus for the TAGs domain and advancing the field's development.
- We propose GraphCLIP, a novel graph-summary pretraining method combined with invariant learning that empowers cross-domain graph foundation models with strong zero-shot capability.
- We introduce a novel graph prompt tuning technique aligned with our pretraining objectives for few-shot settings, mitigating catastrophic forgetting and minimizing learning costs.
- Through extensive experiments, GraphCLIP demonstrates impressive performance in both zero-shot and few-shot scenarios. Furthermore, various downstream tasks are evaluated to validate the universality of GraphCLIP.

## 2 Preliminaries

### 2.1 Notations

In this work, we focus on Text-Attributed Graphs, which incorporate raw text information for each node. Formally, given a text-attributed graph $G = \{\mathcal{V}, \{T_n\}_{n=1}^N, A\}$, where $\mathcal{V}$ denotes the node set with $|\mathcal{V}| = N$ instances, $T_n \in \mathcal{T}^{L_n}$ represents the raw text

for node $n \in [1, 2 \ldots, N]$, $\mathcal{T}$ is the token dictionary, and $L_n$ is the sequence length, $A \in \mathbb{R}^{N \times N}$ denotes the adjacency matrix. To enhance scalability, a sampling function $\Gamma(\cdot)$ is applied to a large graph to derive a set of small ego-graphs $\mathcal{I} = \{G_n\}_{n=1}^N$, where $G_n$ represents the subgraph centered on node $n \in [1, 2 \ldots, N]$.

### 2.2 Problem Definition

To develop a graph foundation model, extensive source graph data $\mathcal{G}^s$ can be utilized to train a general graph model endowed with transferable knowledge:

$$f_{\theta^\star} = \arg \min_{\substack{G_i^s \in \mathcal{I}^s}} \mathbb{E} \, \mathcal{L}_{\text{pretext}} \left( f_\theta; G_i^s \right), \qquad (1)$$

where $\mathcal{I}^s$ represents the set of sampled subgraphs derived from the source data, $f_\theta$ means graph neural networks like GCN [27], GAT [56] and Graph Transformer [45, 70], $\mathcal{L}_{\text{pretext}}$ denotes pretext task like instance discrimination [12]. The optimal pretrained model $f_{\theta^\star}$ can then be applied to low-resource target graph data $\mathcal{G}^t$ to perform downstream tasks such as node classification, link prediction, and graph classification.

In this work, we focus on low-resource settings, including zero-shot and few-shot scenarios, which are critical capabilities for GFMs. For the zero-shot setting, the pretrained GFM can be directly deployed on target data without any adjustment:

$$p_n = \arg \max_n P_{\theta^\star}(\hat{y}_n \mid G_n^t), \quad \forall G_n^t \in \mathcal{I}^t \qquad (2)$$

where $P_{\theta^\star}$ is the pretrained GFM, the prediction for the $n$-th instance is the class with the highest probability. $\mathcal{I}^t$ represents the set of sampled subgraphs derived from the target data.

For the few-shot setting, a limited number of training samples for each class are used for fine-tuning:

$$f_{\theta'} \in \arg \max_\theta \mathbb{E}_{G_n^t \in \mathcal{I}^{t|\text{tr}}} P_\theta(\hat{y}_n = y_n \mid G_n^t), \qquad (3)$$

where $y_n$ is the ground truth of n-th training sample, $\mathcal{I}^{t|\text{tr}}$ represents the training set of sampled subgraphs derived from the target data, $f_{\theta'}$ is the finetuned model which will be evaluated on test samples of target data.

Most existing TAG methods heavily rely on label information from source data in Equation 6, and struggle with low-resource target data, particularly in zero-shot setting. We will present solutions to address these challenges in Section 3.

## 3 Method

In this section, we present our approach to addressing the aforementioned challenges. First, we introduce the technique for generating and curating source data in Section 3.1. Based on this pretraining corpus, we then design a novel contrastive language-graph pretraining method to develop a graph foundation model in Section 3.2. Lastly, we outline the implementation of zero-shot learning on target data and propose a novel graph prompt tuning method for few-shot settings to fully leverage our model's potential in Section 3.3.2. A detailed complexity analysis of GraphCLIP is provided in Appendix F.

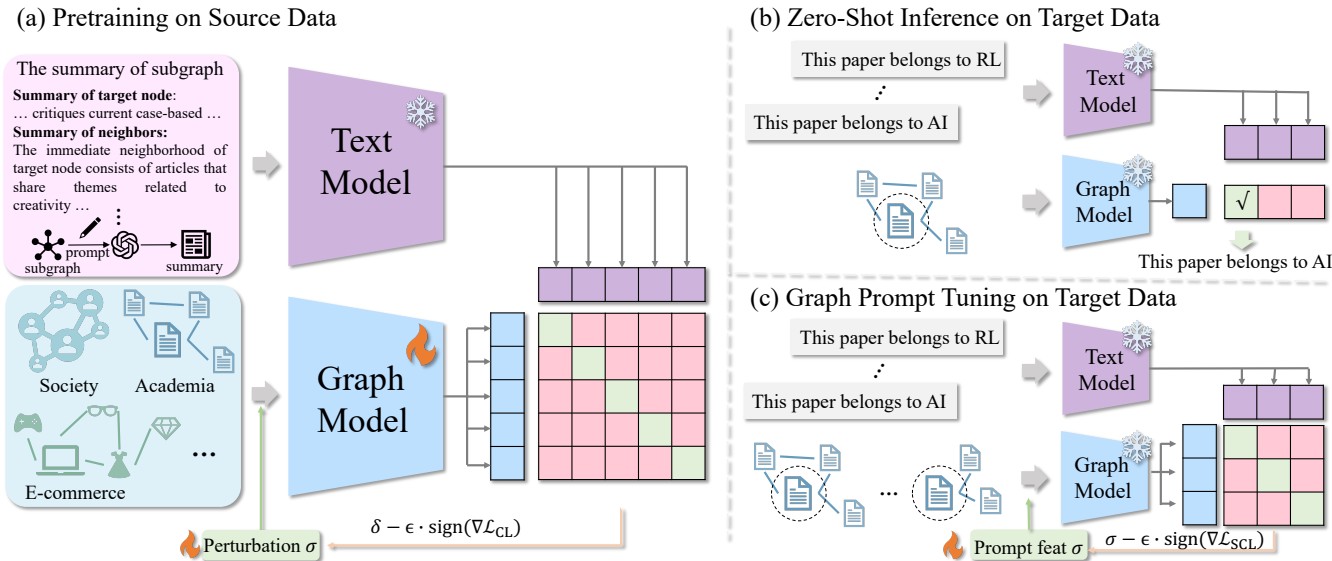

**Figure 2: Our proposed GraphCLIP Framework: (a) represents the self-supervised pretraining method we designed, (b) denotes zero-shot learning of GraphCLIP, and (c) refers to our graph prompt tuning method on target data.**

## 3.1 Graph-Summary Pair Generation

In the TAG domain, there is abundant text describing each node, most current TAG methods leverage this textual information alongside structural data to pretrain graph models; however, these approaches heavily depend on label information [14, 31, 33, 63, 72, 79], restricting scalability due to high labeling costs. Alternatively, some approaches [3, 64] design self-supervised training signals using original text information. Nevertheless, a substantial gap persists between raw textual data and graph-level information, leading to suboptimal performance and limited model transferability. These constraints hinder the development of graph foundation models comparable to CLIP [44] and BLIP [29] in the multimodal domain, which successfully leverage robust self-supervised signals.

To resolve this challenge, we exploit the remarkable summarization capabilities of LLMs to generate pertinent summaries for graphs to construct graph-summary pair data. Specifically, we employ a graph XML-like markup language, such as GraphML [2], and meticulously design prompt templates to enhance the LLMs' comprehension of input graphs, leveraging their adeptness with markup languages [67]. The proposed prompt template for transcribing citation network into markup language is in Template 3.1. In this template, we design two attributes for node content, title and abstract. Additionally, we establish one attribute for describing edge type, *e.g.*, cited, co-purchased or liked. The blue text denotes placeholders that will be replaced with actual data. Using this template, we can seamlessly transform TAG into a format that LLMs can easily comprehend [67].

Considering scalability, we employ a sampling function, random walk with restart sampler, to sample subgraphs $\{G_n\}_{n=1}^N$ from a large graph. These subgraphs will be incorporated into our prompt template and designed instructions to enable LLMs to generate

the corresponding graph summaries $\{S_n\}_{n=1}^N$. In this study, we utilize open LLM, QWen2-72B [69], as the graph summary generator. Combining realistic unlabelled graph data, we generate and curate large-scale graph-summary pair data[1] which contains over 0.2B tokens across academic [14, 58, 71], e-commerce [18], and social domains [19]. For detailed instructions for graph summary generation, please refer to the Appendix C.

```
Graph Prompt Template (Template 3.1)

<?xml version="1.0" encoding="UTF-8"?>
<graphml>
<key id="d0" for="node" attr.name="title" attr.type="string"/>
<key id="d1" for="node" attr.name="abstract" attr.type="string"/>
<key id="d2" for="edge" attr.name="type" attr.type="string"/>
<graph id="G" edgedefault="undirected">
    <node id="n0">
            <data key="d0">{title_0}</data>
            <data key="d1">{abstract_0}</data>
    </node>
    ...
    <node id="nj">
            <data key="d0">{title_j}</data>
            <data key="d1">{abstract_j}</data>
    </node>
    <edge id="e0" source="n0" target="n2">
            <data key="d2" >{relation_0}</data>
    </edge>
    ...
    <edge id="ek" source="ni" target="nj">
            <data key="d2" >{relation_k}</data>
    </edge>
</graph>
</graphml>
```

---
[1]https://anonymous.4open.science/r/GraphCLIP/summary/

## 3.2 Self-Supervised Graph-Summary Contrastive Pretraining

For graph-summary pair data, we employ different encoders to process their information according to their respective modalities. Then, a novel contrastive loss combined with invariant learning is deployed to align these modalities into the same subsapce.

*3.2.1 Graph Encoding.* Considering that model scale is crucial for the emergence and homogenization [34] of graph foundation models, we utilize Graph Transformers (GTs) [45, 66] instead of small GNN models like GCN [27] and GAT [56] to encode graph information. Given a subgraph $G_i = (P_i, X_i, A_i)$, we encode the graph information as follows:

$$h_i = \mathcal{P}(g_\theta(P_i, X_i, A_i)), \qquad (4)$$

where $g_\theta$ denotes the Graph Transformer, such as GraphGPS [45], $P_i$ represents the positional embeddings for the subgraph, *e.g.*, RWPE [7], and $\mathcal{P}$ is the mean pooling function that yields the graph-level encoding $h_i \in \mathbb{R}^{1 \times d}$ for the subgraph.

*3.2.2 Summary Encoding.* To encode sentence or document-level information into a compact and semantically rich vector, we utilize sentence-level text encoders like SBERT [46]:

$$u_i = \mathcal{P}(f_\phi(S_i)) = \mathcal{P}(\text{LM}([\text{CLS}], s_1, s_2, \ldots, s_L)) \qquad (5)$$

where $S_i$ is the summary text for subgraph $G_i$, and $s_1, s_2, \ldots, s_L$ are the tokens of the summary text. where $S_i = [\text{CLS}, s_1, s_2, \ldots, s_L]$ is the summary text of subgraph $G_i$, which is pre-generated by LLMs as described in Section 3.1. We use a mean pooling function $\mathcal{P}$ to average token representations as the summary encoding $u_i$, which captures high-level document-level information. In subsequent sections, $\mathcal{P}$ will be omitted for clarity.

*3.2.3 Contrastive Graph-Summary Pretraining.* After obtaining the graph and summary encodings $H, U \in \mathbb{R}^{N \times d}$, we employ contrastive loss [41] to align the two modalities. Unlike previous multimodal pretraining methods such as CLIP, different graphs can vary significantly across domains, making it essential to capture transferable or causal features in the graph domain. To achieve this goal, we introduce invariant learning [1, 75] efficiently to extract causal features rather than spurious ones. Below we first revisit the concepts of contrastive loss [41] and invariant learning [1]. Then we formulate how they can be combined to solve the challenges of graph foundation models.

DEFINITION 1 (CONTRASTIVE LOSS [41]). *The contrastive loss function is applied to representations, pulling together the positive pairs while pushing apart negative pairs:*

$$\mathcal{L}_{CL}\left(g_\theta, f_\phi; \mathbb{P}, \mathbb{Q}_G, \mathbb{Q}_S, \pi\right) =$$

$$\mathop{\mathbb{E}}_{G,S \sim \mathbb{P}} \mathbb{E}_{\tau_\alpha, \tau_\beta \sim \pi^2} \left\| g_\theta\left(\tau_\alpha\left(G\right)\right) - f_\phi\left(\tau_\beta(S)\right) \right\|^2 \qquad (6)$$

$$- \mathop{\mathbb{E}}_{S \sim \mathbb{Q}_S} \log \mathop{\mathbb{E}}_{G' \sim \mathbb{Q}_G} \mathbb{E}_{\tau' \sim \pi} \left[ e^{\|f_\phi(\tau_\beta(S)) - g_\theta(\tau'(G'))\|^2} \right],$$

*where $G, S \sim \mathbb{P}$ represent positive pairs sampled from the joint distribution of graphs and summaries, while $\mathbb{Q}_G$ and $\mathbb{Q}_S$ denote the marginal distributions of graphs and summaries, respectively. $\tau$ refers*

to the set of data transformations (augmentations) used to generate augmented views. The second line in Equation 6 is termed the alignment loss, and the third line is termed the uniformity loss [60].

However, this loss is not robust to distribution shifts [74, 78] because the expectation operator over different data transformation in the alignment loss can not guarantee the invariant features, resulting poor transferability. Refer to Appendix A for detailed derivation. In order to solve this dilemma, we will combine invariant learning into our method.

DEFINITION 2 (INVARIANT LEARNING [1]). *If a classifier $c_{\omega^*}$ is considered simultaneously optimal for all domains in $\mathcal{H}$, then a data representation $g_\theta$ can elicit an invariant predictor $c_{\omega^*} \circ g_\theta$ across the domain set $\mathcal{H}$:*

$$c_{\omega^*} \in \arg\min_{c_\omega} \mathcal{R}(c_\omega \circ g_\theta; \mathcal{G}) \text{ for all } \mathcal{G} \in \mathcal{H}, \qquad (7)$$

*where $\mathcal{R}$ denotes the risk associated with the predictor $c_\omega \circ g_\theta$ evaluated on the domain $\mathcal{G}$.*

However, this method heavily relies on environment and downstream labels [1], which is not compatible with our self-supervised contrastive loss. To address this issue, we combine the merits of invariant learning and vanilla contrastive loss to obtain a shift-robust contrastive loss, thereby enhancing transferability and generalization across diverse graphs. The core component of our shift-robust contrastive loss is the invariant alignment loss:

DEFINITION 3 (INVARIANT ALIGNMENT LOSS [78]). *The invariant alignment loss $\mathcal{L}_{IAL}$ of the encoders $g_\theta$ and $f_\phi$ over the joint distribution $\mathbb{P}$ of graphs and summaries is defined as follows:*

$$\mathcal{L}_{IAL}(g_\theta; \mathcal{G}) := \mathop{\mathbb{E}}_{G,S \in \mathbb{P}} \sup_{\tau, \tau' \sim \pi} \left\| g_\theta(\tau(G)) - f_\phi\left(\tau'(S)\right) \right\|^2. \qquad (8)$$

The supreme operator quantifies the disparity between two representations under the most "challenging" augmentations, as opposed to the trivial expectation delineated in Equation 6. This methodology enables the invariant alignment loss to generate consistent linear optimal predictors across disparate domains, thus facilitating enhanced out-of-distribution (OOD) generalization and transferability that are deficient in the original alignment loss. Further analysis and theoretical justifications are provided in the Appendix B.

A significant concern regarding the substitution of alignment loss $\mathcal{L}_{AL}$ with $\mathcal{L}_{IAL}$ lies in the impracticality of estimating $\sup_{\tau, \tau' \sim \tau} \|g(\tau(G)) - f(\tau'(S))\|^2$, as this requires iterating through all augmentation spaces. To efficiently identify the worst-case scenario in the continuous space, we employ adversarial training [25, 28, 48, 52, 77, 78] to approximate the supremum operator:

$$\min_\theta \mathbb{E}_{(G,S) \sim \mathbb{P}} \left[ \max_{\|\delta\|_p \leq \epsilon} \mathcal{L}_{CL}\left(g_\theta(X + \delta, A, P), f_\phi(S)\right) \right]. \qquad (9)$$

where the inner loop optimizes the loss to approximate the most challenging perturbation $\delta$, whose magnitude $\|\delta\| \leq \epsilon$ is meticulously regulated to ensure that it does not alter the semantic labels of the original view, *e.g.*, $\epsilon = 1 \times 10^{-2}$. Here, we only add perturbation on graph encoding, because we freeze the text encoder to mitigate catastrophic forgetting and avoid overfitting [40, 49, 80].

## 3.3 Model Adaptation on Target Data

In this section, we introduce the techniques employed to adapt models to target datasets. First, we illustrate the adaptation of GraphCLIP on target data for zero-shot learning. Then, we propose a novel graph prompt tuning method for few-shot learning.

*3.3.1 Zero-shot Learning.* Upon pretraining with Equation 9, our model can be directly deployed on target datasets without any additional training, *i.e.*, enabling zero-shot inference as depicted in Figure 2b. We meticulously craft the prompt to incorporate target label information. For example, in the context of a citation network, the sentence associated with label information is formulated as "This paper belongs to {class}". The specific templates are detailed in the Appendix C. Formally, the formulation of zero-shot learning is as follows:

$$\hat{y}_i = \arg\max_k \mathbb{E}_{u_k} \text{sim}(h_i, u_k), \qquad (10)$$

where sim denotes the cosine similarity function, we identify the most similar label sentence as the predicted label for node $i$.

*3.3.2 Graph Prompt Tuning under Few-shot Setting.* In low-resource scenarios, where only a few samples exist for each class of target data, effectively utilizing this data while preventing over-fitting and catastrophic forgetting [40, 49, 80] becomes crucial. In this work, we introduce a novel graph prompt tuning approach to address this challenge.

Specifically, during the graph prompt tuning process, both the text and graph models remain frozen, allowing only a limited set of parameters to be learnable. To align with our pretraining objective, we incorporate a learnable prompt feature that resemble perturbations used during pretraining, and we employ supervised contrastive loss [24]. The total loss of our designed graph prompt tuning is as follows:

$$\min_\sigma \mathbb{E}_{(G,Z)\sim\mathbb{P}^{\text{tar}}} \left[ \mathcal{L}_{\text{SCL}} \left( g_{\theta^*}(X + \sigma, A, P), f_{\phi^*}(Z) \right) \right], \qquad (11)$$

where $g_{\theta^*}$ and $f_{\phi^*}$ denote the frozen graph and text models, respectively, $Z$ represents the label-related sentence, and $\mathbb{P}^{\text{tar}}$ signifies the distribution of labeled target data. $\mathcal{L}_{\text{SCL}}$ is the supervised contrastive loss [65], which considers pairs with the same labels as positive and those with different labels as negative. After the graph prompt tuning, the evaluation on the target testing data proceeds in the same manner as outlined in Equation 10.

## 4 Experiments

In this section, we first introduce the datasets used in Section 4.1 and the baselines in Section 4.2. We then aim to address the following research questions through our experiments: **RQ1**: How good is the GraphCLIP's in-domain and cross-domain zero-shot transferability? **RQ2**: How effective is our proposed graph tuning in few-shot scenarios? **RQ3**: What impact does the source data have on cross-domain transferability? **RQ4**: What is the effect of hyper-parameters on performance? **RQ5**: How do the main components of our model influence performance?

## 4.1 Datasets

In this work, we utilize 12 open text-attributed graphs across four diverse domains, comprising 5 large-scale TAGs for source data during pretraining and 7 small-scale TAGs for target data evaluation. The statistics of these datasets are detailed in Table 1. To balance the ratio of different source data, we employ the training set of ogbn-Products as pretraining corpus, which consists of around 200K products. More details of these datasets can be found in Appendix D.

|  | #Nodes | #Edges | Domain | #C | Usage |
|---|---|---|---|---|---|
| ogbn-ArXiv [58] | 169,343 | 1,166,243 | Academic | 40 | $\mathcal{G}_{\text{source}}$ |
| ArXiv_2023 [14] | 46,198 | 78,543 | Academic | 40 | $\mathcal{G}_{\text{source}}$ |
| PubMed [71] | 19,717 | 44,338 | Academic | 3 | $\mathcal{G}_{\text{source}}$ |
| ogbn-Products [18] | 2,449,029 | 61,859,140 | E-commerce | 47 | $\mathcal{G}_{\text{source}}$ |
| Reddit [19] | 33,434 | 198,448 | Social | 2 | $\mathcal{G}_{\text{source}}$ |
| Cora [47] | 2,708 | 5,429 | Academic | 7 | $\mathcal{G}_{\text{target}}$ |
| CiteSeer [10] | 3,186 | 4,277 | Academic | 6 | $\mathcal{G}_{\text{target}}$ |
| Ele-Photo [68] | 48,362 | 500,928 | E-commerce | 12 | $\mathcal{G}_{\text{target}}$ |
| Ele-Computers [68] | 87,229 | 721,081 | E-commerce | 10 | $\mathcal{G}_{\text{target}}$ |
| Books-History [68] | 41,551 | 358,574 | E-commerce | 12 | $\mathcal{G}_{\text{target}}$ |
| WikiCS [38] | 11,701 | 215,863 | Wikipedia | 10 | $\mathcal{G}_{\text{target}}$ |
| Instagram [19] | 11,339 | 144,010 | Social | 2 | $\mathcal{G}_{\text{target}}$ |

**Table 1: Statistics of Text-Attributed Graph datasets. $\mathcal{G}_{\text{source}}$ denotes source datasets, and $\mathcal{G}_{\text{target}}$ indicates target datasets.**

## 4.2 Baselines

To evaluate the effectiveness of GraphCLIP, we compare it against 17 baselines, which include: (i) eight LLM-only methods (without modeling structural information of the graphs), *i.e.*, BERT [23], SBERT [46], DeBERTa [13], E5 [59], Qwen2-7B-Insturct [69], Qwen2-72B-Insturct [69], LLaMA3.1-8B-Instruct [55], and LLaMA3.1-Insturct-70B [55], (ii) 4 state-of-the-art TAG methods, *i.e.*, GraphGPT [53], LLaGA [4], OFA [33], and ZeroG [31], and (iii) 5 self-supervised graph algorithms applied to TAGs, *i.e.*, DGI [57], GRACE [81], BGRL [54], GraphMAE [16] and G2P2 [64]. Specifically, to assess the zero-shot performance of discriminative LLMs and self-supervised graph algorithms on target data, we will use the cosine similarity between node embeddings and label embeddings for predictions. For generative LLM methods, we leverage their generative capabilities to estimate their zero-shot performance. Details of these baselines can be found in the Appendix E.

## 4.3 Zero-Shot Inference on Target Data (RQ1)

In order to evaluate the zero-shot transferability of our pretrained model, we conduct experiments of node-level and link-level tasks.

*4.3.1 Node Classification.* In this subsection, we will perform zero-shot node classification by directly applying pretrained models to target datasets.

*Experimental Setup.* For the LLM baselines, we directly apply them to the source data, as they have already been pretrained on extensive corpora, and we find that continuing to pretrain them on our source data deteriorates performance on the target data. For GraphGPT, we utilize their released checkpoint[2] and conduct experiments under our settings. For other methods, we use their

---

[2]https://github.com/HKUDS/GraphGPT

| $\mathcal{G}_{source}$ | Methods | Params | Cora | CiteSeer | WikiCS | Instagram | Ele-Photo | Ele-Computers | Books-History |
|---|---|---|---|---|---|---|---|---|---|
| – | BERT [23] | 110M | 19.56±0.98 | 33.26±2.35 | 29.37±0.00 | 57.02±0.57 | 21.80±0.14 | 13.88±0.29 | 9.95±0.42 |
| – | SBERT [46] | 66M | 54.35±1.26 | 50.47±0.90 | 48.16±0.00 | 48.34±1.23 | 35.96±0.44 | 41.82±0.22 | 30.45±0.19 |
| – | DeBERTa [13] | 184M | 16.42±1.26 | 16.42±1.26 | 15.29±0.00 | 39.81±0.58 | 12.38±0.26 | 10.62±0.15 | 8.70±0.26 |
| – | E5 [59] | 110M | 44.65±0.82 | 42.57±0.54 | 31.49±0.00 | 61.28±0.97 | 35.14±0.28 | 16.54±0.14 | 12.92±0.48 |
| – | Qwen2 [69] | 7B | 61.44±1.29 | 53.57±0.86 | 30.96±0.05 | 39.13±0.78 | 45.55±0.12 | 59.18±0.20 | 23.79±0.34 |
| – | Qwen2 [69] | 72B | 62.18±0.98 | 60.97±0.87 | 37.67±0.10 | 47.70±0.31 | 52.41±0.39 | 60.88±0.30 | 53.56±0.64 |
| – | LLaMA3.1 [55] | 8B | 57.75±1.21 | 53.54±1.71 | 20.39±0.26 | 39.37±1.14 | 34.38±0.25 | 46.98±0.21 | 22.28±0.18 |
| – | LLaMA3.1 [55] | 70B | 65.72±1.24 | 62.79±1.24 | 37.78±0.18 | 43.68±0.52 | 51.26±0.53 | 61.62±0.42 | 53.33±0.55 |
| $X, A, Y$ | GraphGPT [53] | 7B | 23.25±1.45 | 18.04±1.45 | 6.30±0.26 | 45.12±1.16 | 7.62±0.22 | 29.71±0.83 | 15.92±0.14 |
| $X, A, Y$ | LLaGA [4] | 7B | 21.44±0.65 | 16.07±1.15 | 2.65±0.72 | 41.12±0.94 | 6.50±0.53 | 23.10±0.33 | 11.17±0.58 |
| $X, A, Y$ | OFA [33] | 30M | 37.25±1.38 | 29.64±0.19 | 45.52±1.06 | 32.71±0.16 | 33.03±0.64 | 22.09±0.39 | 16.87±0.93 |
| $X, A, Y$ | ZeroG [31] | 66M | 62.32±1.91 | 52.55±1.23 | 54.93±0.06 | 48.97±0.78 | 45.12±0.65 | 56.20±0.35 | 40.74±0.65 |
| $X, A$ | DGI [57] | 128M | 24.03±1.40 | 18.71±1.22 | 18.86±0.25 | 61.42±1.12 | 13.96±0.17 | 27.12±0.03 | 15.77±0.02 |
| $X, A$ | GRACE [81] | 128M | 13.69±1.27 | 22.88±1.49 | 16.07±0.32 | 62.23±0.93 | 10.16±0.13 | 10.94±0.12 | 32.39±0.11 |
| $X, A$ | BGRL [54] | 128M | 20.80±1.06 | 26.50±1.22 | 18.35±0.22 | 61.45±0.82 | 5.21±0.22 | 24.12±0.22 | 16.28±0.35 |
| $X, A$ | GraphMAE [16] | 128M | 23.25±1.07 | 20.75±0.88 | 12.14±0.20 | 62.39±0.84 | 12.53±0.08 | 8.36±0.06 | 21.76±0.17 |
| $X, A$ | G2P2 [64] | 63M | 41.51±0.78 | 51.02±0.62 | 31.92±0.15 | 52.87±0.78 | 22.21±0.12 | 32.52±0.13 | 26.18±0.25 |
| $X, A, S$ | GraphCLIP | 150M | **67.31±1.76** | **63.13±1.13** | **70.19±0.10** | **64.05±0.34** | **53.40±0.64** | **62.04±0.21** | **53.88±0.35** |

**Table 2: Zero-shot inference results for node classification across various target datasets. Boldface indicates the best performance.**

official codes and pretrain models using the source datasets specified in Table 1 before applying them to the target datasets. For data splitting, we use the public split for the WikiCS dataset, while for other datasets, we randomly select 20% as testing samples. We report the mean accuracy along with the standard deviation after five runs with different random seeds.

*Analysis.* From Table 2, we can draw several conclusions. First, generative LLMs demonstrate decent zero-shot performance, particularly LLaMA3.1-70B and Qwen2-72B, attributed to their extensive parameters and training on vast source data. However, these models **struggle to leverage structural information**, resulting in subpar performance on certain target datasets; for instance, LLaMA3.1-70B only achieves 37.67% and 43.68% on the WikiCS and Instagram datasets, respectively.

Second, LLaGA and GraphGPT employ graph instruction tuning to bridge this gap, but they tend to overfit to the source data, leading to **poor generalization**. This is evident as GraphGPT and LLaGA achieve only 6.3% and 2.65% accuracy on the WikiCS dataset, significantly trailing behind other methods.

Third, OFA and ZeroG require label information for pretraining on source data, which results in suboptimal cross-domain transferability due to **mismatched label spaces** and a **failure to capture causal features**. For example, ZeroG only achieves 54.93% accuracy on the WikiCS dataset.

On the contrary, our approach utilizes a self-supervised training task combined with invariant learning to enhance both cross-domain and in-domain transferability. Notably, on the cross-domain dataset WikiCS, our method achieves 70.19% zero-shot accuracy, surpassing ZeroG by over 15% absolute improvement. Similarly, on in-domain datasets like Books-History, GraphCLIP outperforms the previous SoTA method, ZeroG, by over 10% absolute improvement.

Lastly, we observe that four self-supervised graph methods, *i.e.*, DGI, GRACE, BGRL, and GraphMAE, consistently underperform

across most target datasets due to their lack of effective alignment between graph and text modalities, thereby **failing to leverage powerful off-the-shelf text encoders**. Furthermore, G2P2 employs a text encoder as an aligner; however, it relies on raw text as input for the text modality, which **lacks comprehensive graph-level descriptions** and fails to capture invariant features, thereby limiting its transferability.

| Methods | Cora | WikiCS | History |
|---|---|---|---|
| SBERT [46] | 77.76±0.54 | 81.55±0.09 | 87.02±0.07 |
| E5 [59] | 69.57±0.60 | 80.69±0.06 | 69.93±0.07 |
| GRACE [81] | 67.19±0.56 | 69.33±0.06 | 76.90±0.05 |
| GraphMAE [16] | 70.17±0.77 | 73.39±0.04 | 83.26±0.07 |
| OFA [33] | 75.36±0.89 | 89.03±0.09 | 86.47±0.10 |
| ZeroG [31] | 81.20±0.82 | 88.82±0.07 | 92.85±0.12 |
| GraphCLIP | **83.15±0.76** | **92.67±0.10** | **96.04±0.05** |

**Table 3: Zero-shot inference for link prediction of different methods across target datasets.**

*4.3.2 Link Prediction.* In this subsection, we perform zero-shot link prediction by directly applying the pretrained models to the target datasets.

*Experimental Setup.* For link prediction, we report the mean AUC score along with the standard deviation after five runs with different random seeds. Since generative LLM methods produce text outputs, which complicates the extraction of logits or probabilities, we exclude them from this evaluation. For the data splitting, we randomly select 50% as testing samples and employ the same pretrained models as discussed in the previous section.

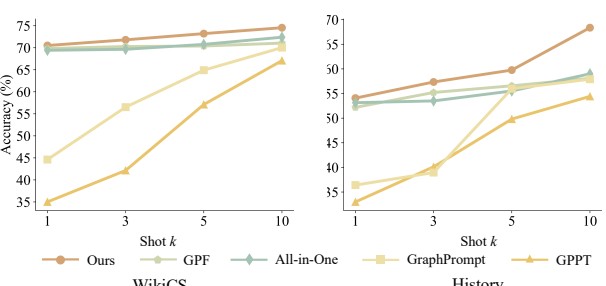

**Figure 3: Node classification of different graph prompt tuning techniques under few-shot setting.**

*Analysis.* From Table 2, we observe that SBERT achieves decent performance. ZeroG, which relies on SBERT and integrates structural information to finetune SBERT through LoRA [17], achieves subpar performance. Notably, our approach demonstrates the best zero-shot performance on link prediction across the evaluated target datasets, highlighting the effectiveness of our designed self-supervised pretraining task and the versatility of our framework.

## 4.4 Graph Prompt Tuning (RQ2)

In low-resource scenarios, where target datasets contain only a few training samples, effectively utilizing these samples while preventing overfitting and catastrophic forgetting is crucial. In this section, we evaluate our graph prompt tuning approach to address this challenge.

*Experimental Setup.* We compare our method with several classical graph prompt tuning methods for node classification, *i.e.*, GPPT [50], GraphPrompt [36], GPF [8], and All-in-One [51]. Each of these methods is applied to our pretrained model, and we evaluate their performance using 1, 3, 5, and 10 shots per class, reporting the mean accuracy.

*Analysis.* From Figure 3, we observe that GPPT and Graph-Prompt fall behind significantly in the 1-shot and 3-shot settings. This underperformance can be attributed to the need for initializing an additional linear head for prediction, preventing direct use of the aligned text model for predictions. However, as the number of shots per class increases, these methods close the performance gap, becoming comparable with others. In contrast, GPF and All-in-One operate directly within the input graph space and can leverage the aligned text model. However, discrepancies between their training objectives and our pretraining objective sometimes result in negative transfer. For instance, in the 1-shot setting, GPF and All-in-One attain accuracy scores of 69.82% and 69.42%, respectively, which are lower than 0-shot performance, *i.e.*, 70.19%.

Our proposed prompt tuning method outperforms the other approaches due to its unified training objectives, effectively mitigating catastrophic forgetting while minimizing the learning cost, leading to superior results.

## 4.5 Explorations on Source Datasets (RQ3)

In this section, we explore the selection of source data for both cross-domain and in-domain transferability. We mask different source

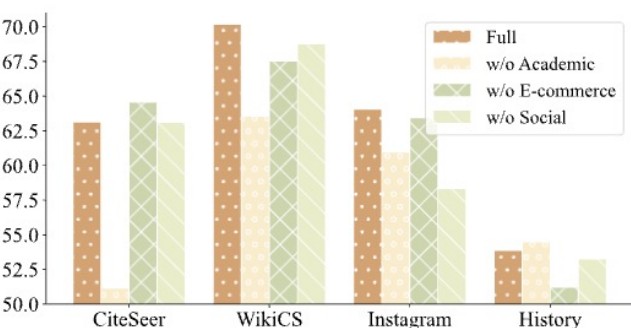

**Figure 4: Analyzing the impact of source data on the performance of target datasets.**

domains and evaluate performance on the CiteSeer (Academic), WikiCS (Wikipedia), History (E-commerce), and Instagram (Social) datasets, as illustrated in Figure 4. The term 'Full' denotes utilizing all source data as described in Table 1. 'w/o Academia' means excluding academic source datasets, *i.e.*, ogbn-ArXiv, ArXiv_2023, and PubMed. 'w/o E-commerce' indicates the exclusion of the e-commerce source dataset, while 'w/o Social' means omitting the Reddit dataset.

From Figure 4, we draw several conclusions. First, a greater amount of source data enhances cross-domain transferability; for instance, 'Full' achieves the best performance on the WikiCS dataset. Second, in-domain source data is critical for in-domain transferability, as demonstrated by 'w/o Academia', which significantly lags behind 'Full' on the CiteSeer dataset, and 'w/o E-commerce', which is inferior to 'Full' on the History dataset. Third, while combining all domains may slightly hurt in-domain transferability, it generally improves overall performance. For example, the performance of 'Full' on the History dataset is slightly lower than 'w/o Academia' but substantially better on the CiteSeer dataset. For simplicity, we use all source data throughout this paper. More complex combinations will be addressed in future work, as balancing the ratio of different source domains remains essential yet challenging.

| Scale | #L | #H | Params | WikiCS | Photo | Computer |
|-------|----|----|--------|--------|-------|----------|
| Small | 4 | 512 | 33M | 67.85 | 50.68 | 57.39 |
| Medium | 8 | 768 | 71M | 69.42 | 51.16 | 58.06 |
| Base | 12 | 1024 | 150M | 70.19 | 53.40 | 62.04 |
| Large | 16 | 1024 | 192M | 70.14 | 54.50 | 60.24 |

**Table 4: Performance of various model scales on target datasets. #L denotes the number of layers, while #H signifies the hidden size.**

## 4.6 Analysis on Model Scale (RQ4)

We explore the impact of model scale in this section. Since the text model remains frozen, our focus is primarily on the scale of the graph model, specifically the graph transformer [45]. We construct four different model scales: Small, Medium, Base and Large. The Small model comprises 4 layers with the hidden size of 512, the Medium model consists of 8 layers with the hidden size of 768, and the Base model, which is used as our primary model throughout

| $\mathcal{G}_{\text{source}}$ | Methods | Cora | CiteSeer | WikiCS | Instagram | Ele-Photo | Ele-Computers | Books-History |
|---|---|---|---|---|---|---|---|---|
| $X, A, S$ | GraphCLIP | 67.31±1.76 | 63.13±1.13 | 70.19±0.10 | 64.05±0.34 | 53.40±0.64 | 62.04±0.21 | 53.88±0.35 |
| $X, A$ | w/o Summary | 58.89±1.77 | 60.67±1.31 | 59.96±0.12 | 59.46±0.62 | 41.65±0.62 | 46.26±0.50 | 44.87±0.35 |
| $X, A, S$ | w/o Freeze | 61.29±1.37 | 61.66±1.01 | 58.92±0.10 | 60.15±0.75 | 45.90±0.65 | 53.24±0.26 | 29.40±0.31 |
| $X, A, S$ | w/o IL | 65.02±1.94 | 60.09±1.09 | 68.64±0.09 | 61.16±0.76 | 51.51±0.60 | 63.21±0.38 | 45.68±0.12 |

**Table 5: Ablation study of masking different components in GraphCLIP.**

this paper, consists of 12 layers with the hidden size of 1024. The Large model has 16 layers with the hidden size of 1024.

From Table 4, we observe that the Base model consistently outperforms the smaller models, likely due to the increased number of parameters [15]. However, while increasing the number of layers to 16 marginally improves performance, it introduces significant computational overhead and, in some cases, even hinders performance on certain target datasets. As a result, we adopt the Base model as our primary model throughout this work.

### 4.7 Ablation Study (RQ5)

In this section, we investigate the impact of different components in GraphCLIP by masking them individually. The term 'w/o Summary' signifies the use of the original text for each node instead of the generated summaries introduced in Section 3.1 as the text modality input. 'w/o Freeze' denotes the non-freezing of the text model, and 'w/o IL' indicates the removal of invariant learning from our pretraining loss.

From Table 5, it is evident that the generated summaries are crucial for achieving zero-shot transferability. The original text, which contains only individual node content, lacks structural information, resulting in a gap between the text and graph modalities. Additionally, original text may include noisy information, leading to suboptimal performance. For example, 'w/o Summary' achieves 41.65% and 46.26% on the Photo and Computers datasets, respectively, falling over 10 absolute percentage points behind GraphCLIP. The 'w/o Freeze' condition shows the poorest performance on the WikiCS and History datasets, with scores of 58.92% and 29.40%. This suggests that fully tuning the text model may lead to overfitting on the source data, impairing transferability. Lastly, 'w/o IL' performs worse than GraphCLIP on most datasets, indicating that incorporating invariant learning into the pretraining loss significantly enhances both cross-domain and in-domain transferability.

## 5 Related Work

### 5.1 Text-Attributed Graph Methods with LLMs

Research on TAGs has gained great attention with rapid development of LLMs, classified into three categories [30]: LLM as Enhancer, LLM as Predictor, and LLM as Aligner, as depicted in Figure 1.

LLM as Enhancer [14, 33, 62, 79] involves augmenting raw text or encoding node features, surpassing traditional methods like Bag of Words (BoW). For example, TAPE [14] uses ChatGPT to enhance node attributes, while OFA [33] and ZeroG [31] utilize language models to unify node features and introduce innovative graph prompting techniques to standardize various tasks. LLM as Predictor [4, 53, 73] uses LLMs to predict graph data by converting it into a comprehensible format. GraphText [73] employs a G-Syntax Tree to transform graph data into text sequences, while GraphGPT and

LLaGA utilize GNNs to encode graph data into tokens, requiring labeled data for training projector but showing limited transferability [5, 32]. LLM as Aligner [3, 42, 64, 72] maps graph and text modalities into a shared embedding space. GLEM [72] optimizes GNNs and LLMs through iterative training, while ConGrat [3] and G2P2 [64] focus on node-text contrastive pretraining, lacking of graph summary text information.

While TAG methods have achieved significant success, they face two primary challenges: (1) heavy reliance on label information, and (2) poor zero/few-shot transferability. In this work, we propose GraphCLIP framework to address these challenges.

### 5.2 Graph Prompt Tuning

In low-resource scenarios, the "pretraining, prompt tuning" paradigm [35] has become a standard approach to address overfitting and catastrophic forgetting. In the graph domain, graph prompt tuning has seen notable success. GPPT [50] introduces task tokens and structure tokens to unify pretraining and downstream tasks such as link prediction. Similarly, GraphPrompt [36] unifies tasks as link prediction, enhancing performance through learnable readout prompt functions. SGL-PT [76] focuses on unifying tasks as masked node prediction, while GPF [8] introduces a learnable universal prompt feature into the input feature for downstream task adaptation. Additionally, All-In-One [51] reformulates all tasks at the graph level by introducing a learnable universal graph prompt, which is inserted into the original graph.

Different to previous studies, our proposed prompt tuning method aims to further align graph and text modalities by leveraging downstream (target) labeled data. Meanwhile, our prompt tuning is unified with the pretraining task, effectively mitigating catastrophic forgetting [40, 49] and minimizing learning costs [50, 76] for superior performance.

## 6 Conclusion

In this work, we propose GraphCLIP, a framework for enhancing the transferability of Graph Foundation Models (GFMs) in low-resource scenarios. Our approach addresses this challenge on two levels. On the data level, we generate and curate a novel, large-scale dataset of graph-summary pairs. On the algorithmic level, we introduce an innovative contrastive graph-summary pretraining method integrated with invariant learning to boost model transferability. Moreover, we develop a novel graph prompt tuning method that aligns with our pretraining task to mitigate catastrophic forgetting. GraphCLIP consistently outperforms existing approaches in both zero-shot and few-shot learning contexts. These results demonstrate the potential for a highly generalizable GFM that can be efficiently and effectively adapted to real-world scenarios.

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

## A    The Dilemma of Vanilla Contrastive Loss

The representation learned through vanilla contrastive loss lacks domain invariance, which is crucial for enabling a model to effectively generalize across diverse target datasets [74, 78]. In this section, we present a specific scenario that highlights the limitations of vanilla contrastive loss. This example, adapted from ArCL [74], demonstrates how encoders trained via vanilla contrastive learning can exhibit markedly different behaviors across varying graph domains, denoted as $\mathcal{G}_\tau$.

PROPOSITION A.1.  *Consider a binary classification scenario with data $(X_1, X_2) \sim \mathcal{N}(0, I_2)$. When $X_1 \geq 0$, the label is set to $Y = 1$, and the data augmentation process involves multiplying $X_2$ by standard normal noise:*

$$\tau_\theta(X) = (X_1, \theta \cdot X_2)$$
$$\theta \sim \mathcal{N}(0,1) \qquad (12)$$

*The resulting transformation-induced domain set is defined as $\mathcal{B} = \{\mathcal{G}_m : \mathcal{G}_m = (X_1, m \cdot X_2) \text{ for } m \in \mathbb{R}\}$. By considering the 0-1 loss, for every $\zeta \geq 0$, there exists a representation $g$ and two domains $\mathcal{G}_m$ and $\mathcal{G}_{m'}$ such that*

$$\mathcal{L}_{AL}(g; \mathcal{G}, \pi) < \zeta \qquad (13)$$

*yet $g$ performs very differently across the domains $\mathcal{G}_m$ and $\mathcal{G}_{m'}$:*

$$|\mathcal{R}(g; \mathcal{G}_m) - \mathcal{R}(g; \mathcal{G}_{m'})| \geq \frac{1}{4} \qquad (14)$$

*where $\mathcal{L}_{AL}$ denotes alignment loss in the vanilla contrastive loss [60], $\mathcal{R}$ denotes supervised risk of binary classification. This example[3] underscores that a representation with a small contrastive loss can still demonstrate significant performance variability across different augmentation-induced domains. The core concept illustrated here is that a low $\mathcal{L}_{AL}$ is achieved by averaging alignments across different augmentation-induced domains, rather than ensuring uniform alignment. As a result, the representation may experience substantial alignment losses in certain less frequently selected domains.*

*Proof.* For $\zeta \geq 0$, let $t = \sqrt{\zeta}/2$ and $g(x_1, x_2) = x1 + tx_2$. Then, the alignment loss of $g$ satisfies:

$$\mathcal{L}_{AL}(g; \mathcal{G}, \pi) = t^2 \mathbb{E} X_2^2 \underset{(\theta_1, \theta_2) \sim \mathcal{N}(0,1)^2}{\mathbb{E}} (\theta_1 - \theta_2)^2 = 2t^2 < \zeta. \quad (15)$$

Set $c$ as 0 and $c'$ as $1/t$, it is obviously that:

$$\mathcal{R}(g; \mathcal{G}_c) = 0 \qquad (16)$$

but

$$\mathcal{R}(g; \mathcal{G}_{c'}) =$$
$$P(X_1 < 0, X_1 + X_2 \geq 0) + P(X_1 \geq 0, X_1 + X_2 \leq 0) = \frac{1}{4} \qquad (17)$$

## B    Theoretical Analysis of Invariant Alignment Loss

In this section, we will give the theoretical justification of why using the supremum operator in invariant alignment loss can addresses the dilemma of original alignment loss. Because it can lower the upper bound of variations across different domains. Formal illustrations are as follow:

---

[3]For simplicity, we assume the adjacency matrix is an identity matrix in this case.

THEOREM B.1 (UPPER BOUND ON VARIATION ACROSS DIFFERENT DOMAINS [74]).  *Given two augmentation functions $\tau$ and $\tau'$, along with a linear predictor $c_\omega$ and representation $g$, the variation across distinct domains is constrained by the following expression:*

$$\sup_{\tau, \tau' \in \mathcal{T}} |\mathcal{R}(c \circ g; \mathcal{G}_\tau) - \mathcal{R}(c \circ g; \mathcal{G}_{\tau'})| \leq m \cdot \|c\| \mathcal{L}_{IAL}(g, \mathcal{G}). \quad (18)$$

*Additionally, fixing $g$ and defining $c_\tau \in \arg\min_c \mathcal{R}(c \circ g, \mathcal{G}_\tau)$, we find that:*

$$|\mathcal{R}(c_\tau \circ g; \mathcal{G}_{\tau'}) - \mathcal{R}(c_{\tau'} \circ g; \mathcal{G}_{\tau'})| \leq$$
$$2m \cdot (\|c_\tau\| + \|c_{\tau'}\|) \mathcal{L}_{IAL}(g, \mathcal{G}). \qquad (19)$$

When $\mathcal{L}_{IAL}$ is minimized to a small value, it signifies that $\mathcal{R}(c \circ g; \mathcal{G}\tau)$ remains consistent across various augmentation functions $\tau$, suggesting that the optimal representation for $\mathcal{G}\tau$ closely resembles that of $\mathcal{G}\tau'$. In other words, representations with smaller $\mathcal{L}_{IAL}$ are more likely to yield similar linear optimal predictors across different domains, a characteristic that the original alignment loss lacks.

## C    Prompt Design

In this section, we present the prompt templates utilized in this study. Initially, we outline the prompt used for generating graph summaries on source data. Then, we provide the prompt template applied for zero-shot learning of baseline models on target data.

### C.1    Prompts for Graph Summary Generation

We present the prompts used for generating graph-summary pair data in Table 6. The violet font represents placeholders, with "seed" to be replaced by the index of the target node within the subgraph. "GraphML" refers to the graph markup language utilized for describing the subgraph, as detailed in Table 3.1. This prompt instructs LLMs to produce both a paper summary and a contextual analysis that encapsulates the essence of the subgraph. We have provided prompts for three source datasets here, as ArXiv_2023 and PubMed are analogous to obgn-ArXiv.

### C.2    Prompts for Baselines

Regarding the prompts for generative LLMs and TAG methods that are based on generative LLMs, we utilize the prompt templates specified in their respective original papers [4, 53]. For illustration, we present the Cora dataset as an example in Table 7. The violet font denotes placeholders, with "raw_text" to be substituted with the original text of the paper. Additionally, "<graph>" signifies graph tokens processed by GNNs.

## D    Datasets

In this section, we present a comprehensive overview of the datasets utilized in this paper. The specifics of **five source datasets** are outlined below:

**ArXiv-2023** dataset, featured in TAPE [14], is a directed graph illustrating the citation network of computer science arXiv papers published in 2023 or later. Similar to OGBN-ArXiv, it consists of nodes representing arXiv papers and directed edges that indicate citations. The objective is to classify each paper into one of 40

| Datasets | Prompts for generating graph summary on source data |
|---|---|
| ogbn-ArXiv | I am providing you with a GraphML file depicting a citation network in computer science. Each node in the network represents a scholarly article, and each edge signifies a citation relationship between articles. Please analyze the article represented by node 'n{seed}' using the provided GraphML data in the following two ways:

1. Paper Summary and Context Analysis:
- Extract and summarize the key findings or contributions of the paper denoted by 'n{seed}'. Consider the details embedded within node 'n{seed}', including its title, abstract, and keywords (if available).
- Provide an overall summary of prevalent themes or concepts shared by the papers that cite or are cited by 'n{seed}' (its direct neighbors in the network). Identify common threads or research topics among these neighbors.

2. Research Area Classification:
- Based on the information summarized from 'n{seed}' and its neighboring nodes, determine the specific research area to which 'n{seed}' primarily contributes.
- Justify the classification by explaining which aspects of 'n{seed}' align with recognized themes, issues, or methodologies in the identified research area(s).

Please ensure your analyses are grounded in the data provided by the GraphML file within 500 tokens, focusing on node 'n{seed}' and its immediate citation neighborhood. The detailed GraphML citation network data is as follows:
{GraphML} |
| ogbn-Products | I have a GraphML file representing an Amazon product co-purchasing network. In this network, nodes represent products sold on Amazon, edges indicate that two products are frequently purchased together. I would like you to analyze the product represented by the node 'n{seed}' using the GraphML data in the following two ways:

1. Product Summary and Context Analysis
- Extract and summarize the details of the product denoted by 'n{seed}', including its title and description (if available).
- Provide an overall summary of the prevalent themes or trends among the products that co-purchased with 'n{seed}'. Identify common threads or topics shared by these neighboring products.

2. Category Classification
- Using the information gathered from 'n{seed}' and its neighboring nodes, classify 'n{seed}' into one of product categories.
- Justify the classification by explaining which aspects of 'n{seed}' align with recognized prevalent themes, trends or threads in the identified product category.

Your analysis should be directly based on the data provided in the GraphML file and should be limited to 500 tokens. Focus exclusively on node 'n{seed}' and its immediate co-purchased neighborhood. The detailed GraphML co-purchased network data is as follows:
{GraphML} |
| Reddit | I have a GraphML file representing a social network where each node denotes a user, the node features are the content of users' historically published subreddits, and edges denote whether two users have replied to each other. I would like you to analyze the user represented by the node 'n{seed}' using the GraphML data in the following two ways:

1. Content Summary and Context Analysis
- Extract and summarize the details of the the user's historical post content denoted by 'n{seed}'. Identify and analyze the user's interests based on their historical posts.
- Provide an overall summary of the prevalent themes or trends among the users that reply with 'n{seed}'. Identify common topics or interests shared by these users.

2. Category Classification
- Using the information gathered from 'n{seed}' and its neighboring nodes, classify whether the user denoted as 'n{seed}' is in the top 50% popular (average score of all subreddits).
- Justify the classification by explaining which aspects of 'n{seed}' align with recognized common topics or interests in the identified user category.

Your analysis should be directly based on the data provided in the GraphML file and should be limited to 500 tokens. Focus exclusively on node 'n{seed}' and its immediate neighborhoods which have replied to each other. The detailed GraphML social network data is as follows:
{GraphML} |

Table 6: Prompts for generating graph-summary pair data.

| Methods | Prompts of baselines for zero-shot learning on target data |
|---|---|
| LLMs | Paper: {raw_text}
Task: Please classify this paper into one of following categories: Case_Based, Genetic_Algorithms, Neural_Networks, Probabilistic_Methods, Reinforcement_Learning, Rule_Learning, Theory. Output the answer without any explanations.
Answer: |
| LLaGA | Given a node-centered graph: <graph>, each node represents a paper, we need to classify the center node into 7 classes: Case_Based, Genetic_Algorithms, Neural_Networks, Probabilistic_Methods, Reinforcement_Learning, Rule_Learning, Theory, please tell me which class the center node belongs to? |
| GraphGPT | Given a citation graph: <graph> where the 0th node is the target paper, and other nodes are its one-hop or multi-hop neighbors, with the following information:
Title: {raw_text}
Abstract: {raw_text}
Question: Classify the target node into one of the following categories: Case_Based, Genetic_Algorithms, Neural_Networks, Probabilistic_Methods, Reinforcement_Learning, Rule_Learning, Theory.
Give the most likely one category of this paper directly. |

**Table 7: Prompts of baselines for zero-shot learning on target datasets.**

subject areas, including cs.AI, cs.LG, and cs.OS, with classifications provided by the authors and arXiv moderators.

**OGBN-ArXiv** dataset represents a directed graph showcasing the citation network among computer science arXiv papers indexed by MAG [58]. Each node signifies an arXiv paper, with directed edges denoting citations. The goal is to categorize papers into one of 40 subject areas such as cs.AI, cs.LG, and cs.OS, with labels manually assigned by authors and arXiv moderators.

**PubMed** [71] dataset comprises three categories: Experimental studies on diabetes mechanisms and therapies, Type 1 Diabetes research focused on autoimmune processes and treatments, and Type 2 Diabetes studies that emphasize insulin resistance and management strategies. Each category addresses distinct facets of diabetes research, contributing to the understanding and treatment of this multifaceted disease.

**OGBN-Products** [18] dataset includes 2 million nodes and 61 million edges, where each node represents an Amazon product, and edges reflect co-purchase relationships. The classification task involves categorizing products into one of 47 top-level categories.

**Reddit** [19] dataset represents a social network where each node corresponds to a user, with node features comprising the content of users' historically published subreddits, and edges indicating whether two users have responded to each other.

We use the full set of OGBN-ArXiv, ArXiv_2023, PubMed, Reddit datasets and training set of OGBN-Products as pretraining data.

The details of **seven target datasets** are as follows:

**Cora** [47] dataset consists of 2,708 scientific publications classified into seven categories: case-based, genetic algorithms, neural networks, probabilistic methods, reinforcement learning, rule learning, and theory. Each paper in the citation network cites or is cited by at least one other paper, resulting in a total of 5,429 edges.

**CiteSeer** [10] dataset encompasses 3,186 scientific publications categorized into six domains: Agents, Machine Learning, Information Retrieval, Database, Human-Computer Interaction, and Artificial Intelligence, with the objective of classifying each paper based on its title and abstract.

**WikiCS** [38] dataset is a Wikipedia-based dataset designed for benchmarking Graph Neural Networks, comprising 10 computer science branches as classes characterized by high connectivity. Node features are extracted from the corresponding article texts[4].

**Instagram** [19] dataset reflects a social network where edges represent following relationships, nodes signify users, and the prediction task involves classifying users as either commercial or regular.

**Ele-Photo** [68] dataset, derived from the Amazon Electronics dataset [39], consists of nodes representing electronic products, with edges denoting frequent co-purchases or co-views. Each node is labeled according to a three-level classification of electronics products. The text attribute for each node comprises the user review with the highest votes, or a randomly selected review if no highly-voted reviews are available. The task is to classify these products into 12 categories.

**Ele-Computer** [68] dataset, also extracted from the Amazon Electronics dataset [39], consists of nodes representing electronic products, with edges indicating frequent co-purchases or co-views. Each node is similarly labeled according to a three-level classification of electronics products. The text attribute for each node is the user review with the most votes, or a randomly selected review if no highly-voted reviews exist. The classification task involves categorizing these products into 10 categories.

**Books-History** [68] dataset is derived from the Amazon-Books dataset, focusing on items labeled as "History." Nodes represent books, while edges indicate frequent co-purchases or co-views between two books. Each node is labeled according to a three-level classification of the book. The title and description of the book itself serve as the text attributes for the nodes. The task is to classify these books into 12 categories.

## E  Baselines

The details of the baselines are outlined below:

---

[4]We obtain the raw texts of each node from https://github.com/pmernyei/wiki-cs-dataset.

**GraphGPT** [53] aligns the graph encoder with natural language semantics through text-graph grounding, integrating the trained encoder with a LLM via a projector. This two-stage instruction tuning enhances the model's ability to perform graph tasks using natural language, facilitating zero-shot transferability.

**LLaGA** [4] employs node-level templates to convert graph data into structured sequences mapped into the token embedding space, enhancing LLM versatility, generalizability, and interpretability when processing graph-structured data.

**OFA** [33] represents all nodes and edges with human-readable texts, encoding them across various domains into a unified space via LLMs. This framework adapts to diverse tasks by incorporating task-specific prompting substructures into the input graph.

**ZeroG** [31] uses a language model to encode node attributes and class descriptions, addressing cross-dataset zero-shot transferability challenges in graph learning through prompt-based subgraph sampling and lightweight fine-tuning strategies.

**DGI** [57] employs a node-graph contrastive method, contrasting node representations with graph representations, where corrupted embeddings and the readout graph representation are treated as negative pairs, while original node representations are considered positive pairs.

**GRACE** [81] focuses on node-node graph contrastive learning, treating representations from the same original node as positive pairs and others as negative pairs.

**BGRL** [54] follows a similar approach to GRACE but omits negative samples, drawing inspiration from BYOL [11].

**GraphMAE** [16] is a masked autoencoder that masks portions of input node attributes before the encoder compresses the masked graph into latent space, with the decoder aiming to reconstruct the masked attributes.

**G2P2** [64] proposes graph-grounded pre-training and prompting to boost low-resource text classification.

## F  Complexity Analysis

In this section, we present the time and space complexity analysis of GraphCLIP. Considering that GraphCLIP is a self-supervised learning framework, we will compare its complexity with other self-supervised graph learning methods.

### F.1  Time Complexity

The primary time overhead arises from three components: the text model, the graph model, optimizing perturbations, and computing the pretraining loss. For simplicity, we assume the layer count and hidden size of the text model are the same as those of the graph model. The time complexity of the graph model is $O(LN^2D + LND^2)$, and similarly, the time complexity of the text model is $O(LN^2D + LND^2)$. The pretraining loss has a time complexity of $O(N^2D)$. For optimizing perturbations, we run the inner loop M times (set to 3 to approximate the max operation in Equation 9) and accumulate gradients for other parameters in the outer loop. Thus, the total time complexity of GraphCLIP is $O(LN^2D + LND^2)$, ignoring smaller terms, which is of the same order as the classical graph self-supervised method GRACE [81], $O(LN^2D)$.

### F.2  Space Complexity

Each layer of the Graph Transformer has a space complexity of $O(ND + D^2)$ for computing queries, keys, and values. The attention score calculation then incurs a space complexity of $O(N^2 + ND)$, while obtaining the hidden states results in a per-layer space complexity of $O(N^2 + ND + D^2)$. Performing these operations across all layers leads to a cumulative space complexity of $O(LN^2 + LND + LD^2)$. Similarly, the text model has a space complexity of $O(LN^2 + LND + LD^2)$. Finally, the contrastive loss adds an additional space complexity of $O(N^2)$. Consequently, the overall space complexity of GraphCLIP amounts to $O(LN^2 + LND + LD^2)$, which is of the same order as the classical graph self-supervised method GRACE, $O(LN^2 + LND + LD^2)$.

## G  Implementation Details of GraphCLIP

In this section, we detail the implementation of GraphCLIP. First, we describe the experimental setup for GraphCLIP during pretraining. Next, we present the prompts utilized in zero-shot learning. Finally, we explain the experimental configurations for prompt tuning.

### G.1  Pretraining Phase

For GraphCLIP, only a few hyperparameters need to be adjusted. In our main experiments, we utilize the AdamW [37] optimizer with both learning rate and weight decay set to 1e-5. The graph model employed is GraphGPS [45], consisting of 12 layers with a hidden size of 1024. For the text model, we use a fine-tuned version of MiniLM [5] [61], featuring 6 layers with a hidden size of 384. To align both models in a unified subspace, a projector is applied to transform the graph model's 1024 dimensions to match the 384 dimensions of the text model. During pretraining, we optimize only the parameters of the graph model and projector, keeping the text model frozen to reduce training costs and mitigate catastrophic forgetting. Pretraining is conducted over 30 epochs with a batch size of 800 per GPU, utilizing eight A100-40G GPUs for pretraining within 7 hours. We will release our pretrained checkpoint [6] after the anonymous phase.

### G.2  Zero-shot Learning

In zero-shot learning, we incorporate label information into label-specific sentences to align with the pretraining format. Table 8 presents various prompts designed for different datasets. Placeholders are marked in violet font: "{class}" represents the label text for the target node, and "{class_desc}" is a descriptive sentence generated by LLMs to elaborate on the label. For detailed cases, please refer to our anonymous repository [7].

### G.3  Graph Prompt Tuning

During prompt tuning, we utilize the AdamW [37] optimizer, with a learning rate of $1 \times 10^{-4}$ and a weight decay of $1 \times 10^{-5}$. The training is conducted over 100 epochs.

---

[5]https://huggingface.co/sentence-transformers/all-MiniLM-L6-v2
[6]https://anonymous.4open.science/r/GraphCLIP/checkpoints/
[7]https://anonymous.4open.science/r/GraphCLIP/data/load.py

| Datasets | Prompts for zero-shot learning of GraphCLIP |
|---|---|
| Cora | "this paper has a topic on {class} {class_desc}" |
| CiteSeer | "good paper of {class} {class_desc}" |
| WikiCS | "it belongs to {class} research area {class_desc}" |
| Instagram | "{class} {class_desc}" |
| Ele-Photo | "this product belongs to {class} {class_desc}" |
| Computers | "is {class} category {class_desc}" |
| History | "this book belongs to {class} {class_desc}" |

**Table 8: Prompts for zero-shot learning of GraphCLIP**

## H  Limitations

In this work, we do not incorporate complex edge attributes, which can be critical for certain graph tasks [18, 20], such as molecule property prediction [18], where each edge may possess distinct properties. Addressing this complexity requires encoding various edge attributes within a unified space and extending Graph Transformers to process these attributes. In future work, we will expand our framework to integrate complex edge information.

