# OpenReview forum: "GraphCLIP: Enhancing Transferability in Graph Foundation Models for Text-Attributed Graphs"
_ACM.org/TheWebConf/2025/Conference — WWW 2025 Poster_

### Official Review · Reviewer_ExV2 · 2024-11-22

**Novelty:** 5
**Technical Quality:** 5

**Review:**

### quality

The paper proposes a novel framework, GraphCLIP, which addresses the challenges of transferability in Graph Foundation Models (GFMs) for Text-Attributed Graphs (TAGs). The authors propose a self-supervised contrastive graph-summary pretraining method that leverages Large Language Models (LLMs) to generate graph-summary pairs, enhancing cross-domain zero/few-shot transferability. The quality of the research is high, as evidenced by the thorough experimentation and comparison against 17 baselines, demonstrating the superiority of GraphCLIP in various settings.
### clarity
The paper is well-structured and clearly written. The introduction effectively sets the stage by outlining the challenges in TAG approaches and the motivation behind GraphCLIP. The methodology is explained in a detailed and logical manner, making it easy to follow. The results are presented clearly, with extensive experiments that validate the effectiveness of the proposed framework.
However, some mathematical formulations (e.g., contrastive and invariant alignment loss) are dense and hard to grasp; more intuitive examples are recommended.
### originality
1. **Graph-Summary Pair Generation**: The idea of using LLMs to generate graph-summary pairs is novel and provides a new way to leverage the power of LLMs in the context of graph learning. This approach addresses the limitations of existing methods that rely on raw text or lack comprehensive graph-level descriptions.
2. **Incorporation of Invariant Learning**: The integration of invariant learning into the contrastive loss is an original contribution. It helps to enhance the transferability and generalization of the model across different graphs, which is a significant improvement over previous methods.
3. **Unified Graph Prompt Tuning**: The proposed graph prompt tuning method that aligns with the pretraining task is also an original aspect of the work. It effectively mitigates catastrophic forgetting and minimizes learning costs, leading to better performance in few-shot scenarios.
### significance
1. **Advancing Graph Foundation Models**: GraphCLIP has the potential to significantly advance the field of graph foundation models by addressing the challenges of transferability. The ability to learn graph models with strong cross-domain zero-shot transferability is highly valuable in real-world applications where labeled data may be scarce or expensive to obtain.
2. **Leveraging LLMs for Graph Learning**: The work demonstrates the effectiveness of leveraging LLMs in graph learning, which is an area of growing interest. By combining the strengths of LLMs and graph neural networks, GraphCLIP opens up new possibilities for developing more powerful graph models.
3. **Providing a New Benchmark**: The generation and curation of a large-scale graph-summary pair dataset contribute to the field by providing a valuable resource for future research. This dataset can be used to evaluate and compare other graph learning methods, facilitating further advancements in the area.
### pros
1. **Good Performance**: The proposed GraphCLIP framework shows strong performance in both zero-shot and few-shot learning scenarios, outperforming existing methods in many cases.
2. **Innovative Approach**: The use of LLMs to generate graph-summary pairs and the incorporation of invariant learning are novel and effective techniques.
3. **Extensive Experiments**: The paper provides a comprehensive set of experiments that demonstrate the effectiveness of GraphCLIP across various datasets and scenarios.
### cons
1. **Complexity of Mathematical Formulations:** Some mathematical concepts may be difficult for readers to grasp.
2. **Insufficient Comparison with Generative Models**: While generative large language models (e.g., LLaMA) are mentioned, the paper does not deeply illustrate the impact of using source data for pre-training LLMs on model performance through specific data comparisons.

**Questions:**

1. **Scalability and Computational Costs**: What are the hardware requirements for pretraining GraphCLIP on the datasets mentioned? How does the computational cost compare with existing methods like ZeroG or G2P2?
2. **Prompt Tuning and Generalization**: Could the proposed graph prompt tuning method generalize to non-TAG tasks or other modalities (e.g., image-text)? If so, would adjustments be necessary?
3. **Alternative Approaches**: How would the performance vary when using alternative invariant learning techniques or different text encoders (e.g., non-LLM-generated summaries)?

**Reviewer Confidence:**

3: The reviewer is confident but not certain that the evaluation is correct

**Scope:**

3: The work is somewhat relevant to the Web and to the track, and is of narrow interest to a sub-community

---

### Official Review · Reviewer_LDqL · 2024-12-01

**Novelty:** 5
**Technical Quality:** 5

**Review:**

**Summary**

This paper proposes the GraphCLIP framework, which addresses two key challenges: heavy reliance on label information and limited cross-domain zero/few-shot transferability. By generating large-scale graph-summary pair data with the assistance of LLMs and introducing a graph-summary pretraining method combined with invariant learning, GraphCLIP learns graph foundation models with strong cross-domain zero/few-shot transferability. Experimental results demonstrate the effectiveness and transferability of GraphCLIP, showing consistent performance improvements over compared baselines.

**Strengths**

1. This paper develops a self-supervised contrastive graph-summary pretraining method combined with invariant learning, enhancing the capabilities of LLMs in understanding and transferring across different domains on text-attributed graphs.
2. Extensive experiments and analyses on 12 open text-attributed graph datasets across four
   diverse domains show the superiority of GraphCLIP in both zero-shot and few-shot settings.
3. The paper is well-written and easy to follow.

**Weaknesses**

1. In Section 3.1, this paper utilizes an open LLM QWen2-72B  to generate large-scale graph-summary pair data with constructed instructions in Table 6. How does the method ensure that hallucination issues are effectively avoided? Additionally, how do the authors evaluate the accuracy and quality of responses from the open LLM, which may struggle with graph data understanding? Inaccurate responses could lead to low-quality training instructions, ultimately affecting model performance on various graph tasks across different domains. Furthermore, this process seems time-consuming and brings additional computational costs due to the large amount of data across various domains (as mentioned in Line 318). Could the authors clarify these aspects?
2. Given the token limitations inherent in LLMs, constructing informative instructions is crucial, especially when dealing with large-scale graphs. While the authors employ a sampling function random walk with restart sampling to sample subgraphs from a large graph (Lines 284-287), this simple strategy fails to ensure the quality and information content of sampled subgraphs. This could lead to missing critical information and introducing noise, which negatively impacts the training data quality and reduces the model’s performance on graph tasks.
3. Could the authors provide specific examples of how GraphCLIP is tested on tasks, such as node classification and link prediction, including task-specific instructions and model's responses? Did the authors input some task-specific instructions (such as node classification and link prediction) as training data during the pre-training stage to help the model better understand these tasks?
4. The meaning of $'X, A, Y', 'X, A', 'X, A, S'$ in Table 2 and Table 5 should be clarified.
5. Related Work (Section 5) should add some studies on graph contrastive learning and invariant learning, which are central to the paper’s method.

**Questions:**

See the above **Weaknesses**.

**Reviewer Confidence:**

4: The reviewer is certain that the evaluation is correct and very familiar with the relevant literature

**Scope:**

4: The work is relevant to the Web and to the track, and is of broad interest to the community

---

### Official Review · Reviewer_2A8e · 2024-12-02

**Novelty:** 6
**Technical Quality:** 6

**Review:**

This paper presents a novel framework, GraphCLIP, aimed at improving the transferability of graph foundation models (GFMs) in zero-shot and few-shot learning scenarios. The authors propose a self-supervised contrastive graph-summary pretraining method combined with invariant learning to address the challenges of heavy reliance on label information and limited cross-domain transferability in current TAG methods. The paper is well-written and provides a comprehensive introduction to the problem, related work, and methodology. The experiments are extensive and demonstrate the effectiveness of GraphCLIP across various datasets and tasks.


Strengths:
1. The proposed GraphCLIP framework is innovative and addresses a significant gap in the current literature on TAGs.
2. The paper includes a wide range of experiments that validate the effectiveness of GraphCLIP in both zero-shot and few-shot settings
3. The paper is well-written and easy to follow.

Weaknesses:
1. The graph summary generation is a crucial process which determines the model performance. I suggest the authors to give some examples of the generated graph summary for illustration, and provide some detailed analysis on the dataset.
2. I am confused about the learnable prompt feature in graph prompt tuning. Is it just a vector or scalar? Is there any other solution for prompt tuning? I suggest evaluating other solutions as well.
3. A case study is required.

**Questions:**

-

**Reviewer Confidence:**

3: The reviewer is confident but not certain that the evaluation is correct

**Scope:**

4: The work is relevant to the Web and to the track, and is of broad interest to the community

---

### Official Review · Reviewer_RT5s · 2024-12-03

**Novelty:** 3
**Technical Quality:** 3

**Review:**

This paper tackles the challenges of transferability in Text-Attributed Graphs (TAGs) by introducing a novel framework called GraphCLIP. The primary goal is to improve the zero-shot and few-shot learning capabilities of graph foundation models (GFMs). The authors highlight two key limitations of existing approaches: (i) a heavy dependence on labeled data, and (ii) limited cross-domain transferability. To address these issues, the authors propose a self-supervised graph summary contrastive pretraining approach and incorporate an invariant alignment loss to enhance performance.

Strength :
1. The paper proposes a novel graph-summary pretraining method that creatively uses large-scale LLM-generated summaries to align subgraph and text modalities. This approach extends the success of multimodal models like CLIP to the graph domain.

2. The introduction of invariant alignment loss demonstrates a deeper understanding of transferability challenges.

3. The framework is versatile and applicable to both zero-shot and few-shot scenarios, making it suitable for low-resource domains.

Weaknesses:
1. The problem definition is unclear, and Equations (1), (2), and (3) do not appear to be aligned. Additionally, the notation for the functions is confusing, particularly the symbols such as $f_{\theta^*}$, $P_{\theta^*}$, $f_{\theta'}$, etc.

2. Although the experiments demonstrate the effectiveness of the proposed method, incorporating the graph transformer model (GraphGPS) makes it unfair to compare against purely text-based methods. It is worth noting that the performance of the proposed method appears weaker than the plain GraphGPS model reported in the benchmarking results.

3. The conclusions drawn from the experiments are not well-supported. For instance, the claim that the method mitigates catastrophic forgetting "due to its unified training objectives" is made without any theoretical or empirical analysis. Simply presenting comparative results is insufficient to demonstrate that the model effectively addresses catastrophic forgetting.

**Questions:**

1. The subgraph extraction process is based on random walks. Could you provide the detailed settings for different graphs? Specifically, the graph depth, walk length, and the number of nodes used for each graph.

2. How the graph neural network is being pretrained, Specifically what is the instance discrimination to tune the graph model in equation 1.

**Reviewer Confidence:**

3: The reviewer is confident but not certain that the evaluation is correct

**Scope:**

4: The work is relevant to the Web and to the track, and is of broad interest to the community